# Prevalence and Determinants of Social Media Addiction among Medical Students in a Selected University in Saudi Arabia: A Cross-Sectional Study

**DOI:** 10.3390/healthcare11101370

**Published:** 2023-05-10

**Authors:** Mansour A. Alfaya, Naif Saud Abdullah, Najim Z. Alshahrani, Amar Abdullah A. Alqahtani, Mohammed R. Algethami, Abdulelah Saeed Y. Al Qahtani, Mohammed A. Aljunaid, Faisal Turki G. Alharbi

**Affiliations:** 1Preventive Medicine and Public Health Resident, Ministry of Health, Abha 62585, Saudi Arabia; 2Consultant of Preventive Medicine and Public Health, Ministry of Health, Abha 62585, Saudi Arabia; 3Department of Family and Community Medicine, University of Jeddah, Jeddah 21589, Saudi Arabia; 4College of Medicine, King Khalid University, Abha 61421, Saudi Arabia; 5Preventive Medicine and Public Health Resident, Ministry of Health, Jeddah 21577, Saudi Arabia; 6Faculty of Medicine, Imam Mohammad Ibn Saud Islamic University, Riyadh 12985, Saudi Arabia

**Keywords:** social media addiction, depression, anxiety, medical students, Saudi Arabia

## Abstract

Social media addiction has become a serious public health concern due to its adverse psychological effects. Therefore, the purpose of this study was to assess the prevalence and determinants of social media addiction among medical students in Saudi Arabia. A cross-sectional study was designed. Participants (*n* = 326) from King Khalid University in Saudi Arabia completed the sociodemographic information, patient health questionnaire-9 scale, and the generalized anxiety disorder-7 tool to measure explanatory variables. The Bergen social media addiction scale (BSMAS) was used to measure social media addiction. A multiple linear regression model was fitted to investigate the predictors of social media addiction. The prevalence of social media addiction among study participants was 55.2% (mean BSMAS score: 16.6). According to the adjusted linear regression, male students had higher social media addiction scores than their female counterparts (β = 4.52, *p* < 0.001). Students’ academic performance was negatively associated with social media addiction scores. Moreover, students with symptoms of depression (β = 1.85, *p* = 0.005) or anxiety (β = 2.79, *p* = 0.003) had a higher BSMAS score compared to their counterparts. Further longitudinal studies are warranted to identify the causal factors of social media addiction, which would assist intervention initiatives by policymakers.

## 1. Introduction

In the twenty-first century, social media (such as Snapchat, Instagram, etc.) has absorbed a significant portion of people’s lives. Social media platforms have sparked a lot of public attention, to the extent that they are now a crucial aspect of modern communication. These means of communication encourage social connection, aid in the upkeep of relationships, and provide a platform for self-expression [1]. People can maintain relationships with existing social network members and create new acquaintances through social media without regard to their location or availability [2,3]. Moreover, social media can be used as an intervention tool to improve health; for example, WhatsApp-based interventions were found to be effective to improve physical activity among female college students in Saudi Arabia [4].

However, excessive and compulsive use of social media sites can lead to an addiction to them, which can have detrimental effects on daily life in multidimensional ways [5]. For instance, some people become so engrossed in Instagram that they become upset when they cannot use it while at work. Such abuse is often referred to as social media addiction [6,7]. Social media addicts sometimes spend too many hours a day on it, become overly preoccupied with it, and have uncontrollable cravings to use the platform [8]. Consequently, addictive or problematic use of social media can impair users’ psychosocial functioning and well-being, and cause symptoms such as depression and anxiety [9,10,11].

Globally, social media addiction has grown to be a major area of public health concern. In 32 countries, the prevalence of social media addiction was recently evaluated by a meta-analysis, which revealed a pooled prevalence of 24% (95% confidence interval: 21% to 28%) [12]. Zhao et al. (2022) [13] conducted an exploratory study among Chinese college students (aged 16 to 23 years) and found that female gender, impulsivity, anxiety, social anxiety, and negative attentional biases were risk factors for social media addiction. Moreover, social media addiction has been associated with individuals’ poor sleep, self-esteem, and body mass index [14,15,16].

In the Kingdom of Saudi Arabia, the use of social media is expanding rapidly. From 7.6 million in 2014 to 29.5 million in 2022, the staggering growth of social media users testifies to the fact that social media platforms are mushrooming in the country [17]. This data indicates that there may be a possibility for the harmful use of social media among Saudi populations. Hence, a robust investigation of social media addiction among vulnerable groups such as young adults, medical students, university students, etc. is urgently needed in the country. In this regard, very few studies [16,18,19] have investigated social media addiction or social media-related outcomes among different populations, including university students in Saudi Arabia. A study conducted among female university students in Saudi Arabia reported that social media addiction has a significant positive association with participants’ body mass index (BMI), but not body image [16]. Another Saudi Arabian study revealed that social networking has a negative effect on dental students’ academic performance [19]. Most of these studies had a small number of independent variables and lacked analytical statistics [16,18]; hence, our study included a wide range of variables such as sociodemographic status, academic performance, physical activity, and mental health (e.g., depression and anxiety), and investigated risk factors of social media addiction using analytical statistical approaches to fill the knowledge gap.

Medical students may suffer from various psychological distress due to their challenging academic obligations [20]. Moreover, students moving from high school to medical college may experience a variety of obstacles and risky behaviors, including addictive behaviors, depression, alexithymia, burnout, and anxiety [21,22]. Today’s medical students are future doctors who will serve society; if they have a social media addiction, it will hinder both their capacity to carry out everyday tasks, and their learning process. Therefore, the purpose of the study was to assess the prevalence and determinants of social media addiction among a sample of medical students sampled from a public university in Saudi Arabia. We hypothesized that sociodemographic and psychological factors would be associated with social media addiction among medical students.

## 2. Materials and Methods

### 2.1. Study Design and Protocol

An institution-based cross-sectional study was conducted among medical students at King Khalid University (KKU) in Abha, Saudi Arabia, to assess the determinants of social media addiction. The survey data were gathered from the participants via an online platform. A Google survey link was emailed to the students, which contained a consent form and a pre-structured questionnaire. The English version of the questionnaire was used since medical students are quite proficient in the English language. The consent form clearly outlined the study’s goals, and participants received guarantees about the privacy and security of their data. Furthermore, it was made clear that participants may discontinue the study at any moment, and taking part in it did not count towards their course requirements. The study design and protocol were reviewed and approved by the Institutional Research Ethics Committee of KKU (approval number: ECM#2023-803). The study design and overall methodological framework are depicted in Figure 1.

### 2.2. Participants, Sample Size, and Sampling

Both male and female medical students from their 2nd academic year to the intern level were eligible to participate in the survey. First-year medical students were excluded from the study as the first year is an introductory year. A single sample population test was used to calculate the sample size. The equation is as follows: Minimum sample size, $n=\frac{z^{2}\;\times \;p\;\times \;(1\;-\;p)}{d^{2}}$.

Here, the following assumptions were taken into account: (i) Due to the limited research on social media addiction in Saudi Arabia, a 50% prevalence was considered (*p* = 0.5). (ii) A 95% level of confidence (Z = 1.96) and (iii) 5% margin of error was considered (d = 0.05). Thus, a minimum required sample of 384 participants was calculated. Unfortunately, the actual sample size for our study, which was 326, was smaller than the anticipated size. We sent out invitations to 450 students to take part in the study; however, 326 students made up the study sample because 17 datasets were missing and 107 students declined to take part.

To recruit the study participants, a systematic recruitment process was used. At first, we managed medical students’ contact details including email addresses from the Deanship of Admissions and Registration of KKU. Participants were selected using a random sampling technique proportionate to the number of medical students at each level. Students were selected using a simple random sampling technique using an Excel sheet that contained the email for each student at each study level. Finally, we sent them (i.e., study participants) an email with an attachment of a Google survey link.

### 2.3. Study Tool and Measure

The study questionnaire contained a total of 37 variables under four segments: (i) sociodemographic and behavioral information (variable = 13), (ii) assessment of depressive symptoms (variable = 09), (iii) assessment of anxiety symptoms (variable = 07), and (iv) social media addiction (variable = 06) and apps-related information (variable = 02). The questions for obtaining sociodemographic and behavioral information were prepared by the study team based on the country’s perspective. Three validated tools, such as the patient health questionnaire (PHQ-9), generalized anxiety disorder (GAD-7) scale, and Bergen social media addiction scale (BSMAS) were used to assess depressive symptoms, anxiety symptoms, and social media addiction, respectively. Approximately 12–15 min were needed to complete the study questionnaire.

#### 2.3.1. Sociodemographic and Behavioral Information

Participants’ sociodemographic and behavioral characteristics including gender, age, marital status, monthly family income, education level, grade point average (GPA), residence, fathers’ education level, mothers’ education level, housemates, current smoking status, self-reported BMI, and physical activity level were obtained.

#### 2.3.2. Assessment of Depressive Symptoms

The PHQ-9 scale, which has nine items, was used to measure depressive symptoms [23]. Each response was coded on the PHQ-9 scale from “Not at all” (0) to “nearly every day” (nearly every day) (3). Each item’s total score, which ranges from 0 to 27, is calculated by adding up all the responses. A score of 10 or more was considered an indication of having depression, while a score of less than 10 was considered to have no depressive symptoms [24]. It is well-established that the PSQI-9 scale is a validated tool for screening depression in Saudi Arabia [25,26]. Moreover, previous epidemiological research conducted among Saudi medical students used this scale [27,28]. The PSQI-9 scale’s internal consistency in the current investigation was good (Cronbach’s alpha = 0.77).

#### 2.3.3. Assessment of Anxiety Symptoms

The GAD-7 tool was used to assess anxiety symptoms among study participants [29]. A previous study from Saudi Arabia evaluated the psychometric validity of the GAD-7 scale and recommended the use of this scale for identifying anxiety symptoms among university students [30]. Seven items make up the GAD-7 scale, and each one is based on a four-point Likert scale, ranging from “Never” (0) to “Nearly every day” (3). A subject can receive a score from 0 to 21, with a higher score indicating more severe anxiety symptoms. In this study, those scoring 10 or above were deemed to have anxiety symptoms [15]. In the current study, an acceptable level of reliability was found for the GAD-7 scale (Cronbach’s alpha = 0.72).

#### 2.3.4. Measurement of Social Media Addiction

Social media addiction was the outcome variable of this study. The six-item BSMAS was used to evaluate social media addiction [31]. This scale, which takes into account salience, mood, modification, tolerance, withdrawal conflict, and relapse, is based on the ‘components model of addiction’ developed by Griffiths [32]. This measure evaluates social media addiction based on symptoms and associated adverse effects brought on by problematic use during the previous year. A Likert scale with five possible responses, from “very rarely” (1) to “very often” (5), was used to answer every question on the BSMAS. The raw scores for each item are added up to produce the final score (score range: 6–30), with a higher score indicating greater degrees of social media addiction. According to the strict polythetic classification scheme, if an individual scored ≥ 4 on at least two-thirds of the total items (i.e., cutoff score = 16 was used), this could be an indicator of social media addiction [12,33]. We only used this classification scheme to estimate the prevalence of social media addiction. In our study, Cronbach’s alpha for this scale was 0.83, which shows high internal consistency.

Moreover, we obtained information about the social media apps (types and frequently used) from the participants by the following two closed-ended questions: (i) Which social media apps do you have on your smartphone? (ii) Which of the following social media apps do you frequently use on your smartphone? There were seven options available: Snapchat, Facebook, Instagram, TikTok, Twitter, WhatsApp, and others.

### 2.4. Statistical Approach

Both descriptive and inferential statistics were performed. Regarding the descriptive statistics, frequencies, percentages, mean, and standard deviation were calculated to summarize the study variables. The distribution of social media addiction score (dependent variable) was tested with the Shapiro-Wilk test, and we found that the distribution was normal (W = 0.9819, *p* = 0.205); therefore, a parametric test was appropriate for the data. Hypothesis testing tests, such as an independent sample t-test and a one-way analysis of variance (ANOVA), were performed to assess the mean difference in the score of social media addiction across all the explanatory variables. An independent sample t-test was performed when explanatory variables had two categories (such as gender, marital status, etc.), and a one-way ANOVA was utilized when explanatory variables had more than two categories (such as age, education level, etc.).

In addition, a multiple linear regression model was fitted to assess the determinants of social media addiction. An adjusted regression model was fitted by considering all explanatory variables to obtain the adjusted estimated effect of the predictors on the outcome variable. The final model fulfilled the assumptions regarding the linear regression model. Multicollinearity among covariates was checked with the variance inflation factor (mean VIF = 1.57). The strength of association was presented using regression coefficients (β) with a 95% confidence interval (CI) and standard error.

A *p*-value of less than 0.05 was set as statistically significant throughout the analysis. Data were analyzed by SPSS (IBM version 23.0, Armonk, NY, USA) and STATA (BE version 17.0, StataCorp, College Station, TX, USA).

## 3. Results

### 3.1. Sample Characteristics

Of 326 samples, more than half of the participants were male (59.8%). The mean age of the sample was 22.91 years (SD: ±1.68). More than two-thirds (68.1%) of the students had a GPA of ≥3.5. Around one-third (32.2%) of participants were classified as having depressive symptoms based on the PHQ-9 scale [mean = 9.11, SD = 6.25]. According to the GAD-7 scale, over a quarter of the participants (28.2%) showed symptoms of anxiety [mean = 8.11, SD = 4.9]. Participants’ sociodemographic, behavioral, and mental health-related characteristics are summarized in Table 1.

### 3.2. Social Media Addiction and Its Determinants

The prevalence of social media addiction among study participants was 55.2% (n = 180), based on the strict polythetic classification scheme of BSMAS (see Figure 2). The mean score for the BSMAS was 16.60 (SD: 5.06). As shown in Table 2, the mean score was significantly varied in relation to participants’ gender (*p* < 0.001), age (*p* = 0.006), GPA (*p* < 0.001), depressive symptoms (*p* < 0.001), and anxiety symptoms (*p* < 0.001).

Adjusted linear regression analysis showed that male students have higher social media addiction compared to their female counterparts (β = 4.52; 95% CI: 3.79, 5.25). Students’ GPA was negatively associated with social media addiction score (i.e., as GPA rises, social media addiction scores decline). Students who were classified as having symptoms of depression (β = 1.85; 95% CI: 1.10, 3.02), and anxiety (β = 2.79; 95% CI: 0.95, 4.62) had higher addiction to social media compared to their counterparts (Table 3).

### 3.3. Social Media Apps-Related Information (Types and Frequently Used)

Snapchat, WhatsApp, and TikTok were the top three reported social networking apps that participants had on their smartphones. WhatsApp, followed by Snapchat, was the most frequently used app among the study participants. However, there was no significant association between the number of social media apps and social media addiction score (*p* > 0.05) (Figure 3).

## 4. Discussion

In the present study, the prevalence of social media addiction among medical students was 55.2% (as per BAMAS). This prevalence is almost close to the finding of a previous study conducted among female Saudi university students, which estimated that half of the students (50.1%) had a moderate level of social media addiction [16]. Although different scales were used for assessing social media addiction, both studies observed a higher prevalence of social media addiction among Saudi students. Moreover, our estimated prevalence of social media addiction is higher than the pooled prevalence (13%, 95% CI: 8–19) with the same classification scheme as a meta-analysis which included 34,798 samples from 32 countries [12]. The sample size may have influenced the results of our study; therefore, future studies should include diverse and country-representative samples to observe the prevalence of social media addiction in Saudi Arabia. The reported high prevalence highlights the need to develop and implement social media-related public health policies and awareness programs, particularly targeting students as well as youth across the country to reduce the burden of social media addiction.

This cross-sectional study also found that sociodemographic characteristics (i.e., being male), academic performance (negative correlation), and psychological conditions (such as depression and anxiety) are associated with social media addiction among medical students in Saudi Arabia.

In our sample, male students had higher social media addiction scores than their female counterparts. This result is consistent with a prior investigation by Alnjadat et al. (2019) [34], which involved medical students at the University of Sharjah in the United Arab Emirates. They found a statistically significant relationship between gender and social media addiction, with male students substantially more addicted to social media than female students [34]. In contrast to our findings, numerous studies from different geographic regions showed that the female gender was associated with a problematic or addictive use of social media [13,15,35]. This discrepancy may be caused by the fact that men use social platforms to make possible friends and find compatible partners with shared interests [36]. Additionally, sociocultural factors may influence our study’s findings. For example, in Saudi Arabia, women must adhere to traditional social values, norms, and cultural standards that may hide their names and personal information from being disclosed or announced using social media. Further qualitative or follow-up research is warranted to better understand how gender affects social media addiction.

Since the GPA system is used in colleges and universities to evaluate a student’s performance and academic development, study participants’ GPA reflect their academic performance. Our study showed a negative correlation between student GPA and social media addiction score, which implies that as social media addiction increases, academic performance declines, and vice-versa. This finding is in line with previous studies [37,38]. Addiction to social media means that students have basic symptoms of addiction [32,35], which might have an adverse impact on their everyday studies and social lives. Academic performance is negatively impacted by the fact that students who are addicted to social media spend less time on academic learning [39], are more cognitively distracted, and have trouble concentrating on learning tasks [40,41]. Furthermore, difficulties in controlling emotions (such as mood) due to social media addiction can disrupt students’ academic studies and performance. There is evidence of mood swings due to social media addiction; for instance, Farooqi et al. (2013) [42] reported that the majority of the studied medical students (715 out of 1000 samples) complained of mood fluctuations in their daily life due to the excessive use of Facebook. Furthermore, excessive social media use or social media addiction is linked to poor sleep quality and sleep disorders [43,44,45], which can make medical students tardy or inattentive in class. However, several studies found no association between social media addiction/usage and academic performance [44,46,47].

Another important finding of our study is that psychological outcomes such as symptoms of depression and anxiety are associated with social media addiction. This finding is also supported by the previous studies conducted among different population groups [46,48,49,50,51]. Indeed, depression has a strong independent association with many addictive behaviors, such as social media addiction [52]. For example, depression is found to be associated with Facebook addiction among Bangladeshi students [50]. This association (i.e., depression and social media addiction) may be explained by pointing out that depressed students may spend more time on social media in an attempt to virtually connect with their friends and peers to overcome loneliness (a symptom of depression) and psychological burdens; however, uncontrolled and excessive use leads to addiction. A recent study among Turkish university students showed a statistically significant positive correlation between aggregate social media scores and loneliness scores [53]. In another sense, medical students who are addicted to social media may be diverted from a healthy lifestyle and put under pressure when they compare themselves to their counterparts, which may exacerbate depressive symptoms. The directionality and causality between depressive symptoms and social media addiction or usage are mixed and unclear in the literature; therefore, a longitudinal study is needed to determine whether excessive use of social media causes depression, or whether depressed people are attracted to social media.

Similarly, this study found a significant positive association between social media addiction and symptoms of anxiety. The finding is aligned with previous research performed among students from different countries [46,54,55,56]. A plausible explanation of this finding is that overusing social media encourages prolonged platform browsing, which could lead to risky behaviors such as sleep deprivation and eventually enhance negative psychological reactions [57]. Social media addiction can also increase students’ enjoyment of their own virtual world by fostering interpersonal and social relationships [46]. However, along with such enjoyment comes thoughts of regret for missing out on these connections or relationships, which could result in a different but no less damaging form of anxiety [58]. Further research is recommended to understand how fear of missing out is associated with social media addiction in vulnerable populations such as young adults, university students, and medical students.

In our study sample, WhatsApp (90%) and Snapchat (84.4%) were the most frequently used apps, which is consistent with a previous study conducted among medical students in Saudi Arabia [44]. This previous study reported that WhatsApp (99.4%) and Snapchat (87.1%) were the most commonly used instant conversation sites among the study subjects [44]. Another study carried out among female university students in Saudi Arabia reported that Snapchat (84.7%) and WhatsApp (77.5%) were the most frequently used social networking sites [16]. WhatsApp allows users to transmit and receive a wide variety of media, including text, images, videos, documents, and locations, as well as voice and video calling. Additionally, it gives users the option to make stories for social media platforms [59]. Students can use social media such as WhatsApp for maintaining existing relationships, as well as for informational and educational motives [60]. Our study did not investigate reasons for social media use; however, Al Saud et al. (2019) [16] in their study reported that 74.6% of the studied students use social networking sites for finding information, 69.1% for keeping in touch with their families and friends, and 57.7% for entertainment. Moreover, in our study, no statistically significant association was found between the number of social media apps and social media addiction. It is established that addiction to multiple social media platforms is associated with adverse health consequences. For example, a previous study from Saudi Arabia showed that students who were addicted to three or more social media sites had a three-times higher risk of very poor sleep quality during the weekend [44]. Moreover, students who used WhatsApp or Snapchat for a greater proportion of the day were more likely to have very poor sleep quality during the weekend compared to their counterparts [44].

### 4.1. Implications for Practice

Based on the study’s findings, we can draw the following implications for the betterment of medical students’ addictive behaviors and psychological well-being: (i) The study findings explored some important and exclusive predictors of social media addiction among medical students in Saudi Arabia, which may contribute substantially to the existing literature for developing theories and coping strategies to reduce social-media-induced problems. Our findings confirm that demographic variation should be considered when developing interventions to reduce social media addiction; for instance, unlike in the previous literature, male medical students were more likely to be addicted in the study sample. (ii) The medical student-aged period, considered as emerging adulthood, is a crucial transition stage of the life cycle from early adolescence to adulthood, and their understanding of the world is still developing [61]. Considering this phenomenon, the negative correlation between academic performance and social media addiction implies that instructors and parents should counsel medical students on how to use social media responsibly in order to avoid addiction and misuse. Moreover, medical students must be conscious of their obligation as future health professionals to society and humanity, and family and academics should support them in understanding the negative consequences of social networking platforms. (iii) As mentioned earlier, medical students’ psychological distress such as depression and anxiety are associated with social media addiction. These findings highlight a need for setting up a “mental health care” center on university premises, where students can easily access mental health services. Moreover, this evidence could assist policymakers and university authorities in designing and implementing intervention programs to promote students’ mental health as well as safe social media use. For example, cognitive behavioral therapy (CBT) would be an effective intervention program to reduce medical students’ social media addiction [62,63].

### 4.2. Strengths and Limitations

The study has some limitations. First, the casual interference between dependent and explanatory variables is limited due to the cross-sectional study design. Second, since study participants were recruited from one university, the outcomes cannot be generalized to the whole country or other similar settings. Third, this study had a smaller sample than the calculated sample size due to missing datasets and unwillingness to participate. Fourth, social desirability and self-reporting biases might occur among respondents. Fifth, this study did not consider the presence of clinically diagnosed mental health conditions such as depression in the recruitment of participants. Despite these limitations, this study has a number of methodological strengths. This study employs stringent methodological and statistical approaches with thorough and repeatable procedures for future exploration in other similar settings. Another positive aspect of this investigation is the use of widely-validated research tools (such as BSMAS, PHQ-9, and GAD-7) to measure the study variables.

## 5. Conclusions

The current study concluded that social media addiction (55.2%) exists among medical students at KKU, Saudi Arabia. This study identified some important factors that are associated with social media addiction among medical students in the country, such as sociodemographic status (being male), academic performance (negative correlation), and psychological factors (such as depression and anxiety). Further follow-up or longitudinal studies that may incorporate diverse and nationally representative samples are warranted to identify the causal factors of social media addiction, which would assist intervention initiatives by policymakers.

## Figures and Tables

**Figure 1 healthcare-11-01370-f001:**
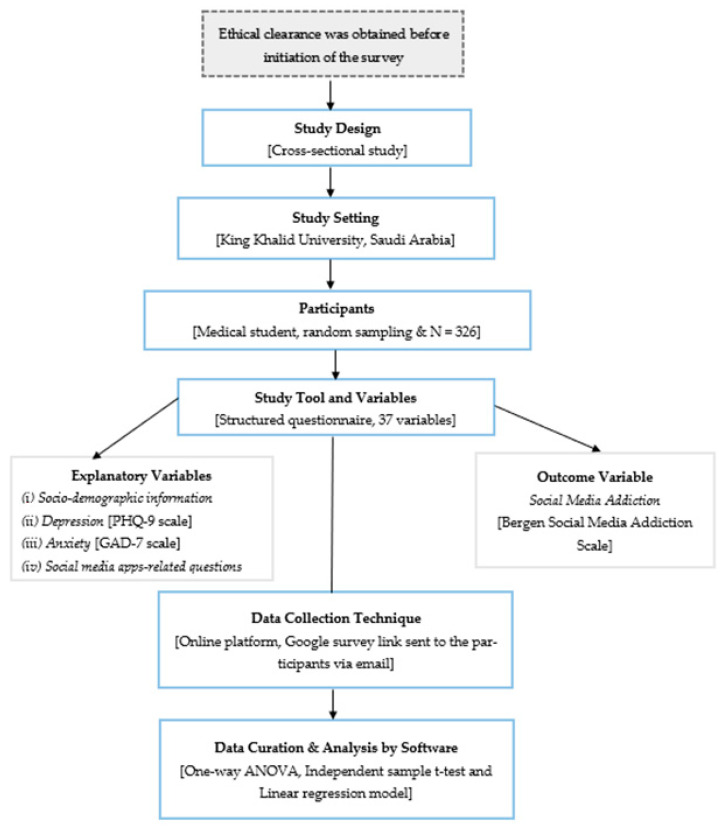
The study design and overall methodological framework.

**Figure 2 healthcare-11-01370-f002:**
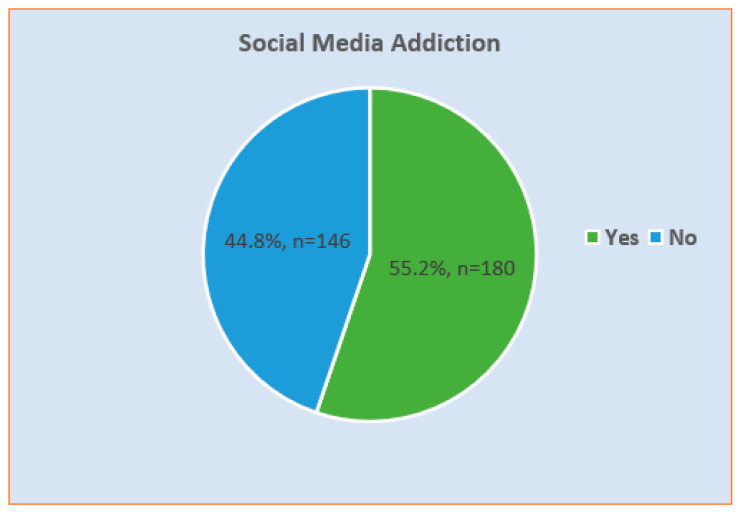
Prevalence of social media addiction among study participants (N = 326).

**Figure 3 healthcare-11-01370-f003:**
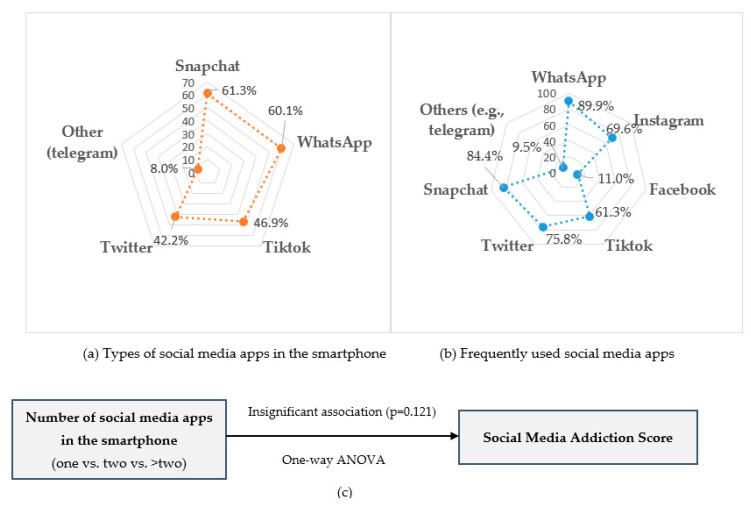
Radar diagram presenting the types (**a**) and frequently used (**b**) social media apps among study participants (*N* = 326). Illustration shows (**c**) the association between number of social media apps in the smartphone and social media addiction.

**Table 1 healthcare-11-01370-t001:** Sociodemographic, behavioral, and mental health-related information among study participants (N = 326).

Variable (s)	Category	Frequency n	Percentage %
Gender	Male	195	59.8
	Female	131	40.2
Age (in years)	18–20	19	5.8
	21–23	186	57.1
	≥24	121	37.1
Marital Status	Single	311	95.4
	Married	15	4.6
Monthly Family Income	<3000 SR	14	4.3
	3000 to 10,000 SR	59	18.1
	10,001 to 20,000 SR	258	48.5
	>20,000 SAR	95	29.1
Education Level	2nd year	13	4.0
	3rd year	49	15.0
	4th year	81	24.8
	5th year	73	22.4
	6th year	64	19.6
	Intern	46	14.1
Grade Point Average (GPA)	<2.5	16	4.9
	2.5–3.49	88	27.0
	3.5–4.5	140	42.9
	>4.5	82	25.2
Residence	Private housing	255	78.2
	Renting housing	52	16.0
	University compound	19	5.8
Educational Level of Father	Illiterate	10	3.1
	Primary level	26	8.0
	Intermediate level	38	11.7
	Secondary level	56	17.2
	University level	196	60.1
Educational Level of Mother	Illiterate	32	9.8
	Primary level	31	9.5
	Intermediate level	35	10.7
	Secondary level	62	19.0
	University level	166	50.9
Living with	Parents	254	77.9
	Alone	46	14.1
	Friends or relatives	26	8.0
Current Smoking Status	Yes	39	12.0
	No	287	88.0
Self-reported BMI Status	Underweight	27	8.3
	Normal weight	147	45.1
	Overweight	85	26.1
	Obesity	87	20.6
Physical Activity Level	No activity	131	40.2
	<3 times per week	108	33.1
	3 to 5 times per week	62	19.0
	>5 times per week	25	7.7
Depression	Yes	105	32.2
	No	221	67.8
Anxiety	Yes	92	28.2
	No	234	71.8

Note: SR = Saudi riyal and 1 USD = 3.75 SR.

**Table 2 healthcare-11-01370-t002:** Differences in social media addiction score shown by explanatory variables (N = 326).

Variable (s)	Social Media Addiction Score			
	Mean	SD	*p* Value from Independent Sample *t*-Test	*p* Value from One-Way ANOVA
Gender			*<0.001 **	
Male	19.30	3.61		-
Female	12.59	4.18		
Age (in years)				*0.006 **
18–20	15.12	4.81	-	
21–23	17.99	5.57		
≥24	14.80	4.79		
Marital Status			0.920	
Married	16.59	5.04		-
Single	16.73	5.71		
Monthly Family Income				0.648
<3000 SAR	15.50	3.92		
3000 to 10,000 SAR	16.13	4.43	-	
10,001 to 20,000 SAR	16.19	5.18		
>20,000 SAR	16.59	5.41		
Education Level				0.898
2nd year	16.00	3.51		
3rd year	16.22	5.38		
4th year	16.85	4.83	-	
5th year	17.04	5.23		
6th year	16.56	5.28		
Intern	16.11	5.08		
Grade Point Average (GPA)				*<0.001 **
<2.5	23.19	5.26		
2.5–3.49	18.65	4.22		
3.5–4.5	17.11	3.81		
>4.5	12.26	4.59		
Residence				0.560
Private housing	16.67	5.25	-	
Renting housing	16.00	4.52		
University compound	17.31	3.79		
Educational Level of Father				0.711
Illiterate	14.40	4.88		
Primary level	16.35	3.51		
Intermediate level	16.89	4.60	-	
Secondary level	16.61	5.31		
University level	16.69	5.27		
Educational Level of Mother				0.543
Illiterate	15.31	5.18		
Primary level	16.16	4.75	-	
Intermediate level	16.94	4.39		
Secondary level	17.13	4.87		
University level	16.67	5.31		
Living with				0.734
Parents	16.59	5.25	-	
Alone	16.98	4.29		
Friends or relatives	16.00	4.59		
Current Smoking Status			0.555	
Yes	16.15	5.18		-
No	16.66	5.06		
Self-reported BMI Status				0.540
Underweight	15.96	6.38		
Normal weight	16.29	5.11		
Overweight	17.16	4.91	-	
Obesity	16.84	4.57		
Physical Activity Level				*0.125*
No activity	17.34	5.66	-	
<3 times per week	16.79	4.36		
3 to 5 times per week	15.47	3.96		
>5 times per week	14.76	6.23		
Depression			*<0.001 **	
Yes	20.90	4.12		-
No	14.56	4.11		
Anxiety			*<0.001 **	
Yes	21.42	3.93		-
No	14.71	4.12		

Note: SR = Saudi riyal and 1 USD = 3.75 SR. Asterisk values indicate statistical significance (i.e., *p* < 0.05).

**Table 3 healthcare-11-01370-t003:** Multiple linear regression analysis showing the determinants of social media addiction among study participants (*N* = 326).

Variable (s)	Adjusted Linear Regression Estimate			
	β	SE	95% CI	*p* Value
Gender				
Male	4.52	0.37	3.79, 5.25	<0.001 *
Female	Reference			
Age (in years)				
18–20	Reference			
21–23	−0.13	0.50	−1.12, 0.81	0.804
≥24	−0.19	0.68	−1.53, 1.15	0.704
Marital Status				
Married	0.88	0.87	−0.83, 2.59	0.312
Single	Reference			
Family Income				
<3000 SAR	Reference			
3000 to 10,000 SAR	1.03	0.89	−0.74, 2.80	0.251
10,001 to 20,000 SAR	1.33	0.85	−0.32, 3.00	0.114
>20,000 SAR	1.37	0.86	−0.33, 3.53	0.211
Education Level				
2nd year	Reference			
3rd year	−1.80	0.94	−3.65, 0.05	0.056
4th year	−1.48	0.92	−3.29, 0.32	0.107
5th year	−1.56	0.99	−3.50, 0.49	0.116
6 year	−1.44	0.99	−3.38, 0.49	0.145
Intern	−2.92	1.07	−5.01, 0.81	0.187
Grade Point Average (GPA)				
<2.5	Reference			
2.5–3.49	−2.34	0.83	−3.97, −0.71	0.005 *
3.5–4.5	−2.63	0.81	−4.22, −1.04	0.001 *
>4.5	−6.15	0.87	−7.86, −4.44	0.001 *
Residence				
Private housing	Reference			
Renting housing	0.09	0.53	−0.95, −0.25	0.863
University compound	1.42	0.85	−0.26, 3.10	0.096
Educational Level of Father				
Illiterate	Reference			
Primary level	0.96	1.17	−1.35, 3.27	0.413
Intermediate level	−0.70	1.15	−2.98, 1.57	0.543
Secondary level	0.15	1.27	−2.06, 2.37	0.892
University level	0.10	1.28	−2.12, 2.32	0.927
Educational Level of Mother				
Illiterate	Reference			
Primary level	0.96	0.81	−0.64, 2.56	0.237
Intermediate level	0.47	0.78	−1.07, 2.01	0.553
Secondary level	0.1.5	0.74	−1.31, 1.61	0.842
University level	0.55	0.69	−0.83, 1.92	0.927
Living with				
Parents	Reference			
Alone	−1.21	0.57	−2.33, 0.27	0.316
Friends, peers and relatives	−2.25	0.71	−2.65, 0.15	0.080
Current Smoking Status				
Yes	−1.25	0.54	−2.31, 0.19	0.222
No	Reference			
Self-reported BMI Status				
Underweight	0.35	0.65	−0.93, 1.63	0.593
Normal weight	Reference			
Overweight	−0.35	0.42	−1.18, 0.47	0.400
Obesity	−0.29	0.44	1.17, 0.57	0.499
Physical Activity Level				
No activity	Reference			
<3 times per week	−0.38	0.42	−1.20, 0.43	0.356
3 to 5 times per week	−1.31	0.49	−2.27, 0.34	0.188
>5 times per week	−1.07	0.67	−2.39, 0.26	0.114
Depression				
Yes	1.85	0.92	1.10, 3.02	0.005 *
No	Reference			
Anxiety				
Yes	2.79	0.93	0.95, 4.62	0.003 *
No	Reference			

Note: β = Regression coefficient, CI = confidence interval. Asterisk values indicate statistical significance (i.e., *p* < 0.05). The R2 for the adjusted regression model was 0.7135.

## Data Availability

Data used in this analysis can be obtained by contacting the corresponding author.

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
