# Peer review of "Prevalence and Determinants of Social Media Addiction among Medical Students in a Selected University in Saudi Arabia: A Cross-Sectional Study"

_healthcare, 2023, doi:10.3390/healthcare11101370_

Round 1

Reviewer 1 Report

This study seems to bring new evidence about the determinants of social media addiction among medical students in Saudi Arabia. Although it uses a cross-sectional design, it demonstrates well standardized sampling strategy and statistical soundness. I would suggest the authors to consider to add some more details (but not mandatory) to make the paper clearer to the readers. 

1. Please consider to add research questions, or hypothesis, in addition to the purpose of the study (line 86). 

2. make a space in "ofmedical" (line 143)

3. While Snapchat and WhatsApps were found to be the most used or installed (Figure 2), no references or discussion has been provided to clarify the influence of each apps, except some ref about Facebook. If possible the authors could discuss popularity of different apps and the reasons.

4. If possible, could the authors consider to add a paragraph in the Discussion to discuss about medical students versus non-medical students.

Author Response

#Reviewer 1

This study seems to bring new evidence about the determinants of social media addiction among medical students in Saudi Arabia. Although it uses a cross-sectional design, it demonstrates well standardized sampling strategy and statistical soundness. I would suggest the authors to consider to add some more details (but not mandatory) to make the paper clearer to the readers. 

  1. Please consider to add research questions, or hypothesis, in addition to the purpose of the study (line 86). 

Authors’ responses: Thank you. We have added a hypothesis at the end of the introduction section. Please read as follows:

“We hypothesized that socio-demographic and psychological factors would be associated with social media addiction among medical students.”

  1. make a space in "ofmedical" (line 143)

Authors’ responses: Corrected.

  1. While Snapchat and WhatsApps were found to be the most used or installed (Figure 2), no references or discussion has been provided to clarify the influence of each apps, except some ref about Facebook. If possible the authors could discuss popularity of different apps and the reasons.

Authors’ responses: Thanks for your suggestion. We have added the following paragraph in the discussion section:

“In our study sample, WhatsApp (90%) and Snapchat (84.4%) were the most frequently used apps, which is consistent with a previous study conducted among medical students in Saudi Arabia [43]. This previous study reported that WhatsApp (99.4%) and Snapchat (87.1%) were the most commonly used instant conversation sites among the study subjects [43]. Another study carried out among female university students in Saudi Arabia reported that Snapchat (84.7%) and WhatsApp (77.5%) were the most frequently used social networking sites [16]. WhatsApp allows users to transmit and receive a wide variety of media, including text, images, videos, documents, and locations, as well as voice and video calling. Additionally, it gives users the option to make stories for social media platforms [58]. Students can use social media such as WhatsApp for maintaining existing relationships as well as for informational and educational motives [59]. Our study did not investigate reasons for social media use; however, Al Saud et al. (2019) [16] in their study reported that 74.6% of the studied students use social networking sites for finding information, 69.1% for keeping in touch with their families and friends, and 57.7% for entertainment. Moreover, in our study, no statistically significant association was found between the number of social media apps and social media addiction. It is established that addiction to multiple social media platforms is associated with adverse health consequences. For example, a previous study from Saudi Arabia showed that students who were addicted to three or more social media sites had a 3-times higher risk of very poor sleep quality at the weekend [43]. Moreover, students who used WhatsApp or Snapchat for a greater proportion of the day were more likely to have very poor sleep quality at the weekend compared to their counterparts [43].

  1. If possible, could the authors consider to add a paragraph in the Discussion to discuss about medical students versus non-medical students.

Authors’ responses: Thanks for your suggestions. However, we didn’t add additional information to discuss medical students versus non-medical students. As you know, comparison is not our study’s objective.  We represent the discussion section based on the objectives of the study. Factors that associated with medical students’ social media addiction is discussed in more in more logical way in the discussion section.  Hope our deny doesn’t make any inconvenience.

Reviewer 2 Report

GENERAL COMMENTS

Nice study. Acceptable for publication after minor changes will have been made.

This reviewer is not qualified enough to evaluate the appropriateness of the statistical analyses used.

SPECIFIC ITEMS

line 18: „Social media addiction ...due to its addictive … effects.” Please don’t use the same descriptive term “addiction … addictive” twice in the same sentence.

lines 28 and 30: This reviewer is unfamiliar with the variable “beta”. How does that translate to a p value? Please give the p value, too.

line 29f: The phrasing is awkward. This reviewer suggests: “Moreover, students with symptoms of depression … or anxiety … had a higher BSMAS score … compared to nondepressed or nonanxious students” – IF that is true: Did they have a higher BSMAS score? If not, how was “addiction” defined in these individuals?

line 50: Please declare – in hours per day – what you mean by “excessive amounts of time”

linge 50: “obsess excessively” is a tautology. “obsess” alone suffices as a description. Again, specify what you mean by “obsess”.

lines 54ff: Please define “social media addiction” before listing prevalence values. IS there a uniform definition of “social media addiction” among the cited references / prevalences?

line 73: “significantly associated” means what? “positively correlated with”? Please specify.

line 80f: That is a bold statement (this reviewer is an MD). IF you want to stick to this statement, compare rates of hospitalizations for psychiatric reasons between medical students and, say, cashiers or construction workers.

lines 80ff: The whole paragraph sounds a little too self-important and whiny. Please tone it down.

line 103: Ethical clearances was “obtained”, not “taken”. Please correct.

line 143: Please correct “ofmedical” to “of medical”.

line 183: “Components model” is a term coined by Griffiths. Please identify as such, eg, “ …. ‘components model’ developed by Griffiths [32].”

line 188: Please correct “06” to “6”.

line 188: Please indicate which BSMAS score is considered to be indicative of social media addiction.

line 195ff: This reviewer is not qualified enough to evaluate the appropriateness of the statistical analyses used.

Table 1, BMI status: Please declare which BMI corresponds to each descriptor.

“Father Education” and “Mother Education” sounds awkward. Suggestion: “Educational level of father” and “…of mother”

Giving percentage values beyond integers is a little absurd. This reviewer takes it that the authors wanted to show that percentages added up to 100. Still …

line 231: The “p” value cannot be zero.Was it smaller than 0.001, perhaps? If so, then write “p < 0.001”.

Table 2: Two significant digits for BSMAS score are enough, e.g., “mean 19, SD 3.6” or, even better “mean 19, SD 4”.

line 236, Figure 2: Please give a legend for figure 2. Please use the same dimensions ie apps in both parts of figure 2 (ie, always draw a heptagram as in the fig 2 b). To explain further, all apps should be in both diagrams and located in the same spot on the heptagram. Please give percentages as integers (e.g., 61% instead of 61.3”) to avoid misleading sophistication (is  “61.3%” really scientifically more meaningful than “61%”?. This reviewer thinks not.)

line 258: You either “explored if” or “found that”. Please clarify.

line 259: You mean “male”, not “female”, correct?

line 274: “We highly recommend” sounds a little pompous. “Further research … is warranted.” sounds more detached.

line 296: sounds awkward. Suggestions: “Symptoms of depression and anxiety are associated with …”

line 301: You do not “justify” this association. Suggestion: “This association may be explained …”

Author Response

# Reviewer 2

GENERAL COMMENTS

Nice study. Acceptable for publication after minor changes will have been made.

 Authors’ responses: Thanks for your appreciation.

SPECIFIC ITEMS

line 18: „Social media addiction ...due to its addictive … effects.” Please don’t use the same descriptive term “addiction … addictive” twice in the same sentence.

Authors’ responses: Corrected. Please read as follows:

“Social media addiction has become a serious public health concern due to its adverse psychological effects.”

lines 28 and 30: This reviewer is unfamiliar with the variable “beta”. How does that translate to a p value? Please give the p value, too.

Authors’ responses: p values are added.

line 29f: The phrasing is awkward. This reviewer suggests: “Moreover, students with symptoms of depression (β = 1.85, p = 0.005), or anxiety (β = 2.79, p = 0.003) had a higher BSMAS score compared to nondepressed or nonanxious students” – IF that is true: Did they have a higher BSMAS score? If not, how was “addiction” defined in these individuals?

Authors’ responses:  We made changes as per your suggestion.

line 50: Please declare – in hours per day – what you mean by “excessive amounts of time”

Authors’ responses:  Corrected.

linge 50: “obsess excessively” is a tautology. “obsess” alone suffices as a description. Again, specify what you mean by “obsess”.

Authors’ responses:  Corrected.

lines 54ff: Please define “social media addiction” before listing prevalence values. IS there a uniform definition of “social media addiction” among the cited references / prevalences?

Authors’ responses:  Revised.

line 73: “significantly associated” means what? “positively correlated with”? Please specify.

 Authors’ responses:  Clarified.

line 80f: That is a bold statement (this reviewer is an MD). IF you want to stick to this statement, compare rates of hospitalizations for psychiatric reasons between medical students and, say, cashiers or construction workers.

 Authors’ responses:  Thank for this comment. We rephrased the sentence. 

lines 80ff: The whole paragraph sounds a little too self-important and whiny. Please tone it down.

 Authors’ responses:  Thanks for your nice observation. We revised this section. Please see the revised manuscript.

line 103: Ethical clearances was “obtained”, not “taken”. Please correct.

 Authors’ responses:  Corrected.

line 143: Please correct “ofmedical” to “of medical”.

 Authors’ responses:  corrected.

line 183: “Components model” is a term coined by Griffiths. Please identify as such, eg, “ …. ‘components model’ developed by Griffiths [32].”

 Authors’ responses:  corrected as per your suggestion.

line 188: Please correct “06” to “6”.

 Authors’ responses:  corrected.

line 188: Please indicate which BSMAS score is considered to be indicative of social media addiction.

 Authors’ responses:  Corrected. Read as follows:

 “According to the strict polythetic classification scheme, if an individual scored ≥4 on at least two-thirds of the total items (i.e., cutoff score = 16 was used), this could be an indicator of social media addiction [12,33]. We only used this classification scheme to estimate the prevalence of social media addiction.” (2.3.4 section)

line 195ff: This reviewer is not qualified enough to evaluate the appropriateness of the statistical analyses used.

 Authors’ responses:  Noted.

Table 1, BMI status: Please declare which BMI corresponds to each descriptor.

Authors’ responses:  Thanks for your nice comment. We measured self-reported BMI status. The questionnaire has only BMI status, no BMI value.

“Father Education” and “Mother Education” sounds awkward. Suggestion: “Educational level of father” and “…of mother”

Authors’ responses:  Corrected as per your suggestions.

Giving percentage values beyond integers is a little absurd. This reviewer takes it that the authors wanted to show that percentages added up to 100. Still …

 Authors’ responses:  You, the reviewer, is right. This commonly used practice and we did it through software.

line 231: The “p” value cannot be zero.Was it smaller than 0.001, perhaps? If so, then write “p < 0.001”.

Authors’ responses:  corrected.

Table 2: Two significant digits for BSMAS score are enough, e.g., “mean 19, SD 3.6” or, even better “mean 19, SD 4”.

Authors’ responses:  Revised.

line 236, Figure 2: Please give a legend for figure 2. Please use the same dimensions ie apps in both parts of figure 2 (ie, always draw a heptagram as in the fig 2 b). To explain further, all apps should be in both diagrams and located in the same spot on the heptagram. Please give percentages as integers (e.g., 61% instead of 61.3”) to avoid misleading sophistication (is  “61.3%” really scientifically more meaningful than “61%”?. This reviewer thinks not.)

Authors’ responses:  Thanks for your nice observation. Your suggestions are good. However, we have drawn this radar diagram indicating the “highest percentage” in the top of the diagram. Hope the presentation would be clear to the readers. We used decimal points for presenting any values as per journal guideline. In this figure, we used one decimal point. 

line 258: You either “explored if” or “found that”. Please clarify.

Authors’ responses:  corrected.

line 259: You mean “male”, not “female”, correct?

 Authors’ responses:  corrected.

line 274: “We highly recommend” sounds a little pompous. “Further research … is warranted.” sounds more detached.

Authors’ responses:  corrected. Please read as below:

“Further qualitative or follow-up research is warranted to better understand how gender affects social media addiction.”

line 296: sounds awkward. Suggestions: “Symptoms of depression and anxiety are associated with …”

 Authors’ responses:  corrected. 

line 301: You do not “justify” this association. Suggestion: “This association may be explained …”

Authors’ responses:  corrected. 

Reviewer 3 Report

The authors discuss an important topic.

Major Issue:

1. please add clarification that the scales were used in English and that students are proficient in English.

Minor issue:

1. need clarification on why the focus is on Medical students. You identified the schooling and performance and nature of education. But has prior lit connected media use with performance--success?

Major strengths:

1. interesting topic and informative

2. Unique to the country and one of the first in the field.

Author Response

# Reviewer 3:

The authors discuss an important topic.

Major Issue:

  1. Please add clarification that the scales were used in English and that students are proficient in English.

Authors’ responses:  Clarified.

“The English version of the questionnaire was used since medical students are quite proficient in the English language.”

Minor issue:

  1. need clarification on why the focus is on Medical students. You identified the schooling and performance and nature of education. But has prior lit connected media use with performance--success?

Authors’ responses:  Thanks for this comment. This is clarified in the introduction section. Please see the revised manuscript.

Major strengths:

  1. interesting topic and informative

Authors’ responses:  Thanks.

  1. Unique to the country and one of the first in the field.

Authors’ responses:  Thanks for your appreciation.

Reviewer 4 Report

Dear authors,

Thank you for coming up with this study: Determinants of Social Media Addiction among Medical students in Saudi Arabia. 

I have few comments for your consideration:

1. Title - since this study is done in only one university, I suggest that you add the phrase - ...in a selected University in Saudi Arabia.

2. I found several sociodemographic profiles such as education of mother and father, family income, residence, etc but which were not discussed in the thesis of the study anywhere in the manuscript aside from being part of the survey and presentation in the results. Why were these included in the profile of the participants? Furthermore, despite the results showing statistically not significant to be among the determinants, it should also be included in the discussion. These profiles were isolated at the middle of the study without any related discussion of its relevance anywhere in the study.

3. What is the difference between university students with dental and medical students?

4. Line 75-76: What is the basis for saying that ALL the studies on Social media addiction done in Saudi Arabia "lacked analytical statistics"? Kindly support this claim.

5. The objectives of the study should include the variables included in the study and as much as possible the hypotheses may also be written. This will give more purposeful focus on the discussion part of the article.

6. Figure 1 may not be necessary

7. How did you rule out presence of medically diagnosed depression among the students? This can be included in the limitation if not done in the recruitment of participants.

8. Study tool and measure - Include in the introduction of this section how many tools were used, estimated time for one participant to complete all tools. 

9. Statistical approach - use descriptive instead of enumerative

10. Results- Subheading for 3.1 should be: Socio-demographic profiles of the participants

11. Figure 2 and its discussion may be move after table 3 and its discussion. Number it as 3.2.3 and write appropriate subheading; while table 1 and its discussion as 3.2.1 and table 3  as 3.2.2 with all its appropriate subheadings

12. What is the cut off  score to determine the presence and level of social media addiction? After which, how many of the participants were considered as social media addicts?  Were all the participants considered as social media addicts? I would like to see this in the results to be presented prior to the inferential analysis results. Add a subsection with a table or graphical presentation of the prevalence of social media addiction and the levels of addiction among the participants.

13. The discussion can be further improved should the hypothesis of the study be articulated.

14. Implications - What is the basis of the inclusion of cultural variation as confirmed results in your study? (Line 332)

15. Implications - Support your claim regarding CBT (Line 349)

16. The conclusion is not consistent with the result- There was no result showing prevalence of social media addiction among the participants; the result showed higher scores of social media addiction among male, and further, the conclusion should be limited to the study setting and not the country.

Thank you.

Several grammatical errors were observed; definitely needs further editing.

Author Response

# Reviewer 4

Dear authors,

Thank you for coming up with this study: Determinants of Social Media Addiction among Medical students in Saudi Arabia. 

I have few comments for your consideration:

  1. Title - since this study is done in only one university, I suggest that you add the phrase - ...in a selected University in Saudi Arabia.

Authors’ responses: Corrected.

  1. I found several sociodemographic profiles such as education of mother and father, family income, residence, etc but which were not discussed in the thesis of the study anywhere in the manuscript aside from being part of the survey and presentation in the results. Why were these included in the profile of the participants? Furthermore, despite the results showing statistically not significant to be among the determinants, it should also be included in the discussion. These profiles were isolated at the middle of the study without any related discussion of its relevance anywhere in the study.

Authors’ responses: Thanks for your thought-provoking comments. We included these sociodemographic profiles as covariates. Inclusion of wide range of variables make the study more rigorous, and provide adjusted estimated result on the outcome variable. We think we should focus on the study’s objectives and hypothesis. As per our study’ objective, we represent the discussion section. This is a common practice in research article to discuss only significant variable and we did follow the fashion. However, we agree that sometimes non-significant variables would be discussed. But, in our case, we got nice findings to contribute the literature.  The revised version, based on editors’ and reviewers comments, improved the manuscript. Thank you again for reviewing our manuscript.

  1. What is the difference between university students with dental and medical students?

Authors’ responses:  Generally, dental students only deal with dental science and dentistry. Basic dental courses are given to the students theoretically by dental teachers. This is followed by practical dental courses, and finally by clinical courses (patient-centered) under the supervision of dental faculty members.. As a result, the dental curriculum is exceptional, and requires dental students to attain diverse proficiencies, including theoretical knowledge, clinical competencies, and interpersonal skills.

The first phase of medical education in Saudi Arabia lasted for over 3 decades. Within this era, the 5 former medical colleges followed the same 6-year traditional curriculum, which consisted of 3 years of basic and medical science courses, 3 years of clinical training, followed by a 1-year internship. There were minor differences between colleges in the arrangement of the subjects and disciplines. Teacher-centred learning strategies were the dominant form of instruction.

Please see the reference for more detail:

https://www.emro.who.int/emhj-volume-17/volume-17-issue-8/article10.html#:~:text=The%20first%20phase%20of%20medical,by%20a%201%2Dyear%20internship.

https://www.saudiembassy.net/education

  1. Line 75-76: What is the basis for saying that ALL the studies on Social media addiction done in Saudi Arabia "lacked analytical statistics"? Kindly support this claim.

Authors’ responses: Corrected and added references.

  1. The objectives of the study should include the variables included in the study and as much as possible the hypotheses may also be written. This will give more purposeful focus on the discussion part of the article.

Authors’ responses: Thank you. We have added a hypothesis at the end of the introduction section. Please read as follows:

“We hypothesized that socio-demographic and psychological factors would be associated with social media addiction among medical students.”

  1. Figure 1 may not be necessary

Authors’ responses: The authors consented to retain the Figure 1. We think this graphical presentation would help readers to understand the study methods at a glance.

  1. How did you rule out presence of medically diagnosed depression among the students? This can be included in the limitation if not done in the recruitment of participants.

Authors’ responses: Included in the limitation section. Please read as follows:

“Fifth, this study did not consider the presence of clinically diagnosed mental health conditions such as depression in the recruitment of participants.”

  1. Study tool and measure - Include in the introduction of this section how many tools were used, estimated time for one participant to complete all tools. 

Authors’ responses: Corrected. Please read as follows:

“The study questionnaire contained a total of 37 variables under four segments: (i) socio-demographic and behavioral information (variable=13), (ii) assessment of depressive symptoms (variable=09), (iii) assessment of anxiety symptoms (variable=07), and (iv) social media addiction (variable=06) and apps-related information (variable=02). The questions for obtaining sociodemographic and behavioral information were prepared by the study team based on the country's perspective. Three-validated tools such as Patient Health Questionnaire (PHQ-9), Generalized Anxiety Disorder (GAD-7) scale, and Bergen Social Media Addiction Scale (BSMAS) were used to assess depressive symptoms, anxiety symptoms, and social media addiction, respectively. Approximately 12-15 minutes were needed to complete the study questionnaire.”

  1. Statistical approach - use descriptive instead of enumerative

Authors’ responses: Corrected.

  1. Results- Subheading for 3.1 should be: Socio-demographic profiles of the participants

Authors’ responses: Corrected. Thanks for your suggestion. As Table 1 is not only cover the participants’ socio-demographic profiles, we change the sub-heading to “sample characteristics”.

  1. Figure 2 and its discussion may be move after table 3 and its discussion. Number it as 3.2.3 and write appropriate subheading; while table 1 and its discussion as 3.2.1 and table 3  as 3.2.2 with all its appropriate subheadings

Authors’ responses: Corrected.

  1. What is the cut off  score to determine the presence and level of social media addiction? After which, how many of the participants were considered as social media addicts?  Were all the participants considered as social media addicts? I would like to see this in the results to be presented prior to the inferential analysis results. Add a subsection with a table or graphical presentation of the prevalence of social media addiction and the levels of addiction among the participants.

Authors’ responses: Thanks for your nice comments. We reported the prevalence of SMA in graphical presentation and discussed it in the discussion section. Please read as below:

“According to the strict polythetic classification scheme, if an individual scored ≥4 on at least two-thirds of the total items (i.e., cutoff score = 16 was used), this could be an indicator of social media addiction [12,33]. We only used this classification scheme to estimate the prevalence of social media addiction.” (2.3.4 section)

Discussion: “In the present study, the prevalence of social media addiction among medical students was 55.2% (as per BAMAS). This prevalence is almost close to the finding of a previous study conducted among female Saudi university students, which estimated that half of the students (50.1%) had a moderate level of social media addiction [16]. Although different scales were used for assessing social media addiction, both studies observed a higher prevalence of social media addiction among Saudi students. Moreover, our estimated prevalence of social media addiction is higher than the pooled prevalence (13%, 95% CI: 8-19) with the same classification scheme in a meta-analysis, which included 34,798 samples from 32 countries [12]. Sample size may have influenced the results of our study; therefore, future studies should include diverse and country-representative samples to observe the prevalence of social media addiction in Saudi Arabia. The reported high prevalence highlights the need to develop and implement social media-related public health policies and awareness programs, particularly targeting students as well as youth across the country, to reduce the burden of social media addiction.”

  1. The discussion can be further improved should the hypothesis of the study be articulated.

Authors’ responses: Thank you. The discussion section is revised based on the reviewers’ comments. Please see the revised manuscript.

  1. Implications - What is the basis of the inclusion of cultural variation as confirmed results in your study? (Line 332)

Authors’ responses: Corrected. We discarded “cultural variation “from the manuscript.

  1. Implications - Support your claim regarding CBT (Line 349)

Authors’ responses: Reference is added.

  1. The conclusion is not consistent with the result- There was no result showing prevalence of social media addiction among the participants; the result showed higher scores of social media addiction among male, and further, the conclusion should be limited to the study setting and not the country.

Authors’ responses: Thank you. Conclusion is revised.

Round 2

Reviewer 4 Report

Dear authors,

Thank you for incorporating the previous suggestions. I just have few minor comments:

1. University students vs medical students- Thank you for extensively differentiating the dental and medical students. However, what I wanted to say was  that medical and dental students are also university students. So, the term university students to describe non-medical or non-dental students will mean that medical and dental students are not students in the university. (Line 71)

2.  Lines 78-79- "that social media addiction was 78 significantly associated with participants’ body mass index (positive correlation),"  can be rephrased to:   that social media addiction has significant positive association with participants’ body mass index...

3. Lines 104 and 106 - change permission to consent

4. If you opt to retain figure 1. I suggest to switch places the outcome and explanatory variables  to ( Right to Left)n the explanatory and outcome variables.

5. Line 217- enumerative to descriptive

6. Line 291- male students were positively associated with higher social media addiction scores compared to their female….. better to say “males have higher addiction…”

7. Grammar issues:

examples- People’s life- people’s lives . Line 42

              Line 139 Participants, sample size and sampling to Line 139                                               Participants, sample size, and sampling

               etc.

Thank you and good luck.

Needs some minor grammar editing.

Author Response

Dear author,

Thank you for incorporating the previous suggestions. I just have few minor comments:

  1. University students vs medical students- Thank you for extensively differentiating the dental and medical students. However, what I wanted to say was  that medical and dental students are also university students. So, the term university students to describe non-medical or non-dental students will mean that medical and dental students are not students in the university. (Line 71)

Authors’ responses: Thank you so much for your nice clarification. Corrected.

  1. Lines 78-79- "that social media addiction was 78 significantly associated with participants’ body mass index (positive correlation),"  can be rephrased to:   that social media addiction has significant positive association with participants’ body mass index...

Authors’ responses: Corrected. Please read as follows:

“A study conducted among female university students in Saudi Arabia reported that social media addiction has significant positive association with participants’ body mass index (BMI), not body image [16].”

  1. Lines 104 and 106 - change permission to consent

Authors’ responses: Corrected.

  1. If you opt to retain figure 1. I suggest to switch places the outcome and explanatory variables  to ( Right to Left)n the explanatory and outcome variables.

Authors’ responses: Corrected.

  1. Line 217- enumerative to descriptive

Authors’ responses: Corrected.

  1. Line 291- male students were positively associated with higher social media addiction scores compared to their female….. better to say “males have higher addiction…”

Authors’ responses: Corrected.

  1. Grammar issues:

examples- People’s life- people’s lives . Line 42

Authors’ responses: Corrected.

Line 139 Participants, sample size and sampling to Line  Participants, sample size, and sampling etc.

Authors’ responses: Corrected.